# Perspectives and Attitudes of Newer New Jersey High School Teachers towards Cleaning, Sanitizing, and Disinfecting Consumer Products Used in School Classrooms

**DOI:** 10.3390/ijerph21020211

**Published:** 2024-02-10

**Authors:** Juhi Aggarwal, Maryanne L. Campbell, Midhat Rehman, Kimberly T. Nguyen, Derek G. Shendell

**Affiliations:** 1NJ Safe Schools Program, Rutgers School of Public Health (SPH), Rutgers University, Piscataway, NJ 08854, USA; mlf159@sph.rutgers.edu (M.L.C.); mr1638@sph.rutgers.edu (M.R.);; 2Department of Environmental and Occupational Health and Justice, Rutgers School of Public Health (SPH), Piscataway, NJ 08854, USA; 3Environmental and Occupational Health Sciences Institute, Rutgers University, Piscataway, NJ 08854, USA

**Keywords:** occupational health and safety, work-based learning, cleaning products, disinfectant products, sanitization products

## Abstract

During the COVID-19 pandemic, there was an increased reported use of chemical cleaning, sanitizing, and disinfecting products (CSDPs), which created public concerns about negative health consequences for both children and adults in public schools. A subset of newer teachers shared experiences regarding safety and health (S&H) while working in school-based settings through a series of online surveys. Surveys were provided to teachers who completed work-based learning supervisory trainings provided by the New Jersey Safe Schools Program between October 2021 and June 2023. The participants answered questions focusing on CSDPs purchased for school use, their attitudes towards CSDPs, their use of personal protective equipment, and symptoms employees may have had due to CSDPs. A total of 205 teacher participants successfully completed the surveys. Over 25% of the teachers did not know where their CSDPs originated from, as they were provided by the school. Most participants “sometimes”, “not often”, or “never” read labels for CSDP ingredients or looked them up on healthy product apps. The participants (60%) tended to wear gloves while cleaning/disinfecting but did not wear masks. A third of the participants experienced respiratory health problems after working at school. Overall, the data suggest that more education on S&H regarding CSDPs needs to be provided to New Jersey teachers.

## 1. Introduction

Due to the COVID-19 pandemic, there was an increase of cleaning, sanitizing, and disinfecting products (CSDPs) used in homes, primary and secondary schools, and universities/colleges [1,2,3]. A 2020 literature review reported that there was no significant transmission of COVID-19 through inanimate surfaces but agreed people should use disinfectant products [4]. In general, CSDPs have health risks when exposure occurs, i.e., if inhaled or if they get on the skin [1,5,6,7,8,9,10,11,12,13]. These products, whether volatized or aerosolized, can be considered indoor air pollutants [14]. Air pollution indoors and outdoors, such as the numerous volatile organic chemical compounds found in CSDPs, can cause severe health problems for both adults and children [15]. In schools, exposures among students—versus adult teachers and/or educational support professionals—to air pollution can cause more damage, since they inhale larger concentrations of pollutants in the air by size and body weight [16]. Poor ventilation has also been linked to lower student academic performances [17].

Cleaners can contain irritants such as acids (acetic acid, diluted hydrochloric acid, etc.), alkalis (ammonia, sodium carbonate, etc.), and bleaching agents (chlorine bleach, hydrogen peroxide, etc.) [18]. During the COVID-19 pandemic, one study noted that United States (U.S.) households reported an increased use of bleach (70.4%) and surface disinfectants (69.7%) [1]. By examining safety data sheets (SDSs), one study discovered that approximately 75% of Swiss manufactured professional cleaning products contained irritants, 64% contained harmful ingredients, and 28% were labeled as corrosive substances [5]. In the general U.S. population, a recent study found that about one-half or 47% of participants reported problems related to the use of cleaning products indoors. The two most commonly reported problems were skin disturbances (68%) and shortness of breath (23%) [1]. U.S. laws [19] via the United Nations [20] require product safety, and potential risks are detailed in SDSs.

The chemicals in both cleaning and disinfecting products can contain many irritants, particularly to the eyes, skin, respiratory tract, and digestive tract [5,6]. Health care workers who were around or used disinfecting products more often reported higher rates of work-related wheezing and watery eyes than non-users, and a nearly three-fold higher rate of asthma than the general U.S. population [10]. Many other studies have also found that the use of disinfecting products led to higher rates of asthma symptom episodes [7,8,9,10], and disinfecting product use increased during the COVID-19 pandemic [10,21].

During the COVID-19 pandemic, there was an increase in the use of CSDPs, especially in schools [2]. While financial support for school infrastructure has been provided by the American Rescue Plan of 2021 [22], teachers and school administrators are often left to buy CSDPs in their community, with little to no guidance. School staff must then determine which CSDPs are best for the health and safety of both the students and the educational professionals in the classroom [23].

Studies have reported that wearing personal protective equipment (PPE) is protective against chemicals found in ingredients in some CSDPs. Henn et al. reported about 9% of participants did not always wear gloves when using disinfectants, while Humann et al. reported that the proportion of time when gloves were used increased when chemical products were used, although this change varied based on occupation [24,25,26]. Another study reported that about 80% of teachers did not wear gloves when cleaning or disinfecting their classrooms [2].

A study in 2020 suggested that grocery store employee(s) in direct contact with customers were five times more likely to contract COVID-19 than employees who did not have direct contact [27]. As teachers come back to in-person schooling, to avoid contracting COVID-19 due to close contact with students and other school personnel, they might attempt to clean classrooms more, leading to higher rates of CSDP use [1,2,3]. During the 2021–2022 and 2022–2023 school years, the New Jersey (NJ) Safe Schools Program (NJSS) asked newer NJ secondary school teachers who completed state-required work-based learning (WBL) supervisory teacher/administrator courses with the NJSS to answer online safety and health (S&H) surveys, including questions about where they buy CSDPs for their classrooms, habits and behaviors when buying then using CSDPs, and any adverse reactions they may have experienced due to exposures to chemicals—known to be hazardous, asthma triggers, or skin irritants—in CSDPs. To our knowledge, there have been few studies to date regarding schoolteachers and CSDPs. This is one of the first studies to specifically survey teachers regarding where they obtain their CSDPs, and their use of PPE while using these items. The purpose of this research is to help ensure teacher and student S&H, given everyone needs to be aware of the products used in their learning/study, living, and workspaces along with possible side effects of exposure to emissions of the products.

The present study aims to determine the “where, what, and how/who” of purchasing supplies, by or for newer public secondary or high school teachers, for cleaning, sanitizing, and disinfecting commonly used on high-contact surfaces throughout the State of NJ during two of three school years (2021–2023) impacted by the COVID-19 pandemic. Also, it aims to determine if education during the first of three school years (2020–2021) impacted by the COVID-19 pandemic helped inform how teachers used available CSDPs during two school years (2021–2023) of an ongoing COVID-19 pandemic. The results from this study can provide guidance to school districts to address the gaps and disconnect in education in practice when it comes to cleaning workplaces in schools and can inform school districts for policies which support teachers and other professionals when it comes to CSDP use and procurement.

## 2. Materials and Methods

### 2.1. Study Population

During the 2021–2022 and 2022–2023 school years, the NJSS provided 163 teachers with WBL supervisory trainings through a special opportunity in collaboration with the NJ Department of Education—the Office of Career Readiness. The goal of this special opportunity was to provide free WBL trainings to teachers who hold a career and technical education (CTE) certificate in NJ via targeted recruitment. Teachers were asked to submit an application form titled “Work-Based Learning Supervisor Trainings for Participating CTE Teacher Application Form” in PsychData, (Psych Data, LLC, State College, PA, USA) to the NJSS. The application was shared statewide to potentially eligible teachers via the NJSS monthly e-newsletter via Constant Contact (Constant Contact Inc., Waltham, MA, USA). Eligibility meant a teacher was endorsed (certification credentials) in CTE, agricultural education, business, human services–cosmetology, allied health, and/or family and consumer sciences, along with being relatively new to CTE and having no more than 10–15 years of overall K–12 teaching experience (ideally < 10 in secondary schools). The NJSS reviewed applications received weekly or biweekly during the school year (given holidays, etc.). Applications were exported from PsychData and managed in Microsoft Excel (Microsoft, Redmond, WA, USA). Those who were eligible received a web link and an approval code for course registration. This special offer was at a discounted price, i.e., only an administrative fee of USD 20 versus the ~USD 750 ± USD 10 cost of the WBL trainings offered by the NJSS during the 2020–2021 and 2022–2023 school years (which those who were not eligible were referred to). This training consisted of six classes highlighting child labor laws and occupational S&H topics for teachers to consider while supervising their working students and a virtual live session with course trainers. After the completion of the training courses and obtaining informed consent through PsychData, the teachers were asked by email to complete either two or three S&H surveys using the same platform. Surveys one and two were provided at the same time a week after the completion of each cohort of trainings via email. Those who took the trainings during the 2021–2022 school year were given a follow-up survey in fall 2022 in the initial month of the 2022–2023 school year. Survey results from all three surveys were combined for one data analysis. Those who completed at least one survey had the opportunity to request a USD 10 e-gift card via email. The researchers could not link email addresses to survey submissions, and survey responses were de-identified and analyzed in aggregate. There was no way to connect participants between surveys due to not collecting identifying information. A total of 205 of 436 (163 for survey 1, 163 for survey 2, and 110 for the follow-up survey) possible entries were received and were either fully or partially completed for a response rate of 47.0%. Our sample size estimate was 55 participants per survey, for a total a sample size of 165 among the three surveys. The estimated sample size for one survey was 55 with a 95% confidence level, a 5% margin of error, and a 4.5% population proportion of CTE teachers in the general teacher population in the New York, NJ, and Pennsylvania metropolitan areas in 2022 [28].

### 2.2. Survey Details

The first survey consisted of questions regarding the following: built or physical school environments, perceptions on S&H, S&H protocols and training, physical hazard concerns, attitudes towards CSDPs, and products purchased for school use. The second survey consisted of questions about the following: where/when teachers worked, any symptoms employees may have experienced, the S&H of the employee, ventilation, awareness of government resources, use of PPE, trainings, personal nutrition and sleep hygiene, and personal physical health and mental health. The follow-up survey consisted of a combination of both surveys. [Please note that the surveys can be found in the Appendix A]. Most questions in these surveys were adapted from previous federal surveys (e.g., the U.S. Census) as used in the NJSS surveys and prior NJSS research surveys crafted by the NJSS to target questions known to be important to teachers [29,30,31]. This paper focuses on survey questions pertaining to products purchased for school use, attitudes towards CSDPs, the use of CSDPs and PPE, and self-reported symptoms teachers experienced due to CSDPs.

### 2.3. Data Analysis

After initial data management and descriptive statistics were computed for the entire population for each question/variable, the data were stratified by gender, race/ethnicity, county of work, age, and the year the WBL trainings were taken with the NJSS. Most (20 of 21) counties in NJ were represented in this study’s sample and were grouped as seven each into North (N), Central (C), and South (S) NJ. Age consisted of two categorical variables, younger than 42 and 42 and above; this was justified since 42 is the current average age for teachers in NJ [32]. Categorical variables were summarized using percentages and were compared between groups using Fisher’s exact test due to small sample sizes. Missing data were excluded from analyses. Calculated *p*-values below 0.05 were considered statistically significant. Data analyses were conducted using Microsoft Excel and the SAS analytics software 9.4 (Cary, NC, USA).

This study was approved by the Institutional Review Board (IRB, or Ethics Committee) of Rutgers, the State University of New Jersey (IRB protocol code: 2021001559).

## 3. Results

### 3.1. Demographics

Overall, across three surveys, a total of 205 entries were received and were either fully or partially completed. For those who chose to answer, 41.9% of the participants identified as male and 58.1% identified as female, which is close to the national average of CTE teachers in the U.S. [33]. Over two-thirds (67.3%) of the participants identified as white, with 59.1% of the overall study population identifying as non-Hispanic white (NHW). Over two-fifths (41.9%) of the participants taught in N.NJ, 29.9% taught in C.NJ or statewide, and 28.1% taught in S.NJ. Most teachers’ (60.9%) highest level of education or degree earned was a master’s degree, 27.2% had a bachelor’s degree, 4.3% had a doctoral degree, and 7.6% had another degree; the average was 6.2 years (Standard Deviation (SD): 2.8) of secondary education. The teachers worked, on average, for 12.6 years in NJ (SD: 7.4) and 13.2 years overall (SD: 7.8). The average birth year for teachers was reported as 1976 (SD: 9.5) (Table 1).

### 3.2. Purchasing Cleaning, Sanitizing, and Disinfecting Consumer Products Habits

The participants were asked where they purchased CSDPs for their classrooms. This study found that 26.9% of the participants did not know where the cleaning products were bought, as the school provided the products; this was also the case for sanitizing (29.0%) and disinfecting (28.8%) products. Most participants (~58.0%) bought their CSDPs at grocery stores, ~28.0% bought CSDPs at big-box stores such as Costco, Sam’s Club, Target, Walmart, etc., and ~12.0% of the participants bought CSDPs by mail, from pharmacies, or hardware stores (Table 2). There were no differences between groups when stratified by gender and race. There were few differences when stratified by other groups.

This study found that a higher percentage of participants who took the trainings in the 2022–2023 school year were more likely to buy cleaning products at a hardware store compared to those who took the survey in 2021–2022 (22.2% in 2022–2023 vs. 4.7% in 2021–2022, *p* = 0.03). The survey also found people who took the trainings in the 2022–2023 school year were more likely to buy disinfecting products by mail compared to those who took the trainings in 2021–2022 (27.8% in 2022–2023 vs. 8.1% in 2021–2022, *p* = 0.03). This study suggests a disparity in age and where people buy their CSDPs. Those who were age 41 and younger were more likely to buy products via mail or parcel services when compared to those 42 and older (Table 3). When stratified by the NJ county of work, this study found that those working in N.NJ were not likely to buy cleaning products in hardware stores, while 13.6–17.4% of the participants in C.NJ and S.NJ reported doing so (*p* = 0.01). Regarding buying cleaning products, no one in C.NJ bought cleaning products online, while between 12.0 and 27.3% of the participants in N.NJ and S.NJ reported doing so (*p* = 0.02). Regarding buying disinfecting products, only one participant in C.NJ bought these products online, while 10.0–27.3% of the participants in N.NJ and S.NJ reported doing so (*p* = 0.05).

### 3.3. Self-Reported Practices of Reading Product Ingredient Lists

The participants were asked if they read the ingredients label in general on their CSDPs. About one-third each read the labels “very often/often”, “sometimes”, or “not often/never” (Table 4). There were no significant differences between any of the assessed stratifications except for race. Those who were NHW were more likely to not read the ingredients on CSDPs compared to those who are not NHW (all *p* ≤ 0.001) (Table 5).

The participants were asked if they read the labels on CSDPs to see if they are made with natural, non-toxic, or eco-friendly “green” ingredients. About one-third read the labels “very often/often”, “sometimes”, or “not often/never” (Table 4 and Table 5). There were no differences when stratified by any of the assessed factors. 

When participants were asked if they look up CSDPs on a healthy product app or website while shopping, over 80.0% of the participants did not use these electronic resources (Table 6). There were no differences when stratified by any of the assessed factors.

When the participants were asked if they needed medical care after using CSDPs, only two participants working in C.NJ secondary schools/districts required medical care: one because of using only cleaning and disinfecting products, and one due to using all CSDPs.

This study found that PPE use while performing cleaning and/or disinfecting services varied depending on the type of PPE, with 60.0% of participants using gloves but only 40.5% using a mask (Table 7). No difference in PPE use while using CSDPs was found between the year the trainings were taken, the NJ county of work, or gender. Those who were not NHW (46.2%) were more likely to wear a mask while using cleaning and disinfecting products than those who were NHW (22.4%) (*p* = 0.02).

When the participants were asked if they used PPE in the 2020–2021 school year, this study found that 79.0% of the participants used PPE, and in the 2021–2022 school year, 73% of the participants did so. Of this 73.0%, 82.2% wore a mask, 49.3% used gloves, and 10.9% used protective eyewear. Those who took the trainings in the 2021–2022 school year were more likely to wear masks at work than those who completed the trainings in 2022–2023 (88.7% in 2021–2022 vs. 45.5% in 2022–2023, *p* = 0.003), but no difference was found in the use of gloves (50.0% in 2021–2022 vs. 45.5% in 2022–2023, *p* = 1.00). Men were more likely to wear protective eyewear when compared to women (50.0% male vs. 20% female, *p* = 0.02).

When the participants were asked if they had any of the following respiratory health problems (runny nose, itchy/watery eyes, trouble breathing, or headache) after working in their school, 31.1% indicated that they did, with 27.3% of those participants seeking medical attention after the event. Only 15.7% of the participants experienced one or more of these symptoms before working in the school, and these symptoms disappeared for 37.3% of the participants if they were out of the workplace for more than a day (Table 7). There were no differences when stratified by any factors.

## 4. Discussion

This study provides valuable insight regarding the habits, behaviors, and preferences of participating NJ secondary school teachers pertaining to the use of CSDPs and PPE in school. The current study found that younger teachers were more likely to buy CSDPs online. This may be because those who are younger have a higher likelihood to use and buy products online in general. During the COVID-19 pandemic, younger participants were more likely to use online health and social care services [34] or online grocery services [35] compared to older participants.

This study also found that about a third of the participants experienced respiratory problems after working in their school, and about a third were relieved of symptoms, i.e., symptoms went away after leaving the school premises. This speculatively could be consistent with the idea that CSDPs are irritating to the respiratory tract, and after leaving the area in which these products are used, one can experience symptom relief [5,6,15,24]. This is not consistent with sick building syndrome, where “most” people in a building will feel ill and “most” will feel better after leaving the building [36]. To our knowledge, this is among the first studies which can be used to compare health-related symptoms reported by teachers within and outside of public secondary schools, before or during the COVID-19 pandemic, and we believe it is the first in the U.S. 

This study found that between those who took the survey in the 2021–2022 school year versus the 2022–2023 school year, there was a decrease in mask use in schools. This may be due to the lift of the mask mandate in NJ schools in March 2022, which was near the start of the last quarter of the 2021–2022 school year [37]. There were no major federal mask mandates in schools at this time. There were not many differences in reported experiences between the 2021–2022 and 2022–2023 school year about participants’ practices regarding the purchasing of CSDPs. The only slight change was that those in the 2022–2023 school year were more likely to buy cleaning products from hardware stores and disinfecting products online when compared to those from the 2021–2022 school year.

This study found that those who are not NHW are less likely to wear masks while using cleaning and/or disinfecting products than those who identified as NHW. A study found that those who did not identify as white males were less likely to have access to respiratory protective equipment that properly fit their face [38]. Another study found that there were differences between those who identified as black and a minority ethnicity and those who did not regarding PPE perceptions. Those who identified as black and minority ethnicities were less likely to feel as if PPE was readily available for them [39]. A study in China reported that there were issues providing PPE to migrant workers at the beginning of the pandemic. This study suggested that potential ethnic discrimination was present, and the lack of PPE usage was not due to a lack of want to use PPE but a lack of access to proper PPE [40]. In this study, it is possible that a lack of access might be another reason NJ CTE teachers who do not identify as NHW do not use PPE as often as those who do. This study also found that men were more likely to wear protective eyewear than women. A study found that women often have more poorly fitting PPE, including goggles, when compared to men [41]. This may be a reason for not using protective eye protection. This paper is thus consistent with patterns seen in other sectors of the workforce.

Due to the fact that many educators do not read the ingredients or look products up on healthy living applications, we suggest that teachers become more educated on CSDPs and their proper use via training. The best time for these trainings to be completed would be prior to the start of each school year, before teachers and students are in the school on a regular basis.

Throughout the COVID-19 pandemic, the NJSS published many papers regarding actions and perceptions among teachers’ responses to different COVID-19-related issues. These studies focused on vaccine practices [42,43], teacher mental health [44], perceptions of WBL activities during the pandemic [45], trends in COVID-19 outbreaks [46], and apprenticeships in NJ [47]. These studies are consistent with the NJSS mission to continue understanding NJ experiences and needs during the pandemic and to ensure safe schools for teachers and students. The NJSS has also offered various S&H trainings online and also disseminated pertinent state and federal COVID-19-related and cleaning/sanitizing/disinfecting workplace resources to schools via monthly e-newsletters and on the NJSS website [48].

This study has strengths and limitations. One strength is the method in which the surveys were distributed. Since this was an online survey, data were all collected, stored, managed, and analyzed digitally. The participants were able to complete the survey at their own pace. The participants’ responses to questions also remained anonymous, and they did not have to worry about repercussions for honest responses submitted to the survey questions and were thus more likely to be open about their opinions. While email addresses were collected for e-gift card distribution, no survey responses were linked to the emails.

Among the limitations of this study, one is that this population is specific, i.e., NJ secondary school CTE teachers, and we had a relatively small sample size of 205 across two school years. The data cannot necessarily be generalized to a broader population outside of NJ, or at least outside of secondary schools and CTE districts. In broader contexts, such as with a larger sample size, research may reveal that what is significant in this study may not continue to be and vice versa. We had a specific sample population, so it is possible that other populations such as teachers in general or even CTE teachers outside of NJ have different perceptions and attitudes regarding CSDPs. These data, however, can be used as a comparison to other states or time frames regarding data for CTE teachers and their perceptions and attitudes regarding CSDPs. Another limitation is due to anonymity; we cannot determine if multiple people took surveys from the same secondary school/district computer available to teachers in a staff room or library and/or if the participants took the survey multiple times. Seven participants completed the surveys multiple times due to the e-gift card opportunity (as per a request from the NJSS to provide email addresses to send it, if the optional incentive was selected); however, it cannot be definitively determined which responses were theirs. Thus, all responses were included in this study. We also cannot determine if a participant took every survey offered to them.

## 5. Conclusions

This study of 205 secondary school career–technical–vocational education or CTE teachers in the state of New Jersey found that over a quarter of its participants used cleaning, sanitizing, and disinfecting chemical-based consumer products or CSDPs provided to them by their school, and over half of the participants bought them at grocery stores. This study did find a difference in the CSDP shopping habits of different age groups, which may potentially be extrapolated to different chemical-containing consumer products or may suggest where teachers buy similar consumer products for use at home. This study also determined that over a third of the participants never read ingredient labels or search for eco-friendly “green” CSDPs, and that less than a fifth look at apps or websites to determine product safety and potential risks via safety data sheets or SDSs, as required by law in the U.S. and via the United Nations. There were also several disparities in the use of personal protective equipment or PPE among different demographic groups by race/ethnicity and gender identity but not age group.

Future research with a larger population sample to better represent the general public K–12 teachers and not only newer CTE teachers is warranted. Another study that focuses on PPE use and potential racial and ethnic differences is also warranted. This study also suggests that more public environmental education is needed on potentially dangerous chemicals found in CSDPs and on the promotion of resources such as healthy product apps or websites from trusted non-profit, university, and government agency sources. Training on PPE usage while using CSDPs in school classrooms and other workplaces to avoid potential negative adverse health effects or symptoms of chronic diseases like asthma is also required. More studies need to be conducted in schools, which would allow researchers to look at the long-term impacts of using CSDPs in classrooms or if different classroom environments (typical classroom, labs, workshops, and salons) experience different problems regarding CSDPs. This could also include an examination of if teacher behaviors change towards CSDP selection and use after learning more about the dangerous ingredients (active and/or inactive chemicals) in CSDPs. The results from this study can also provide guidance for school districts for creating new policies to protect teachers and other school professionals. One policy we suggest is that schools buy CSDPs for teachers and check that these products are safe and have clean ingredients, along with providing PPE for teachers to use as they clean their classrooms. Another related policy is to actually require teachers to use PPE while cleaning their classrooms including proper gloves, masks, and eyewear. A third policy recommendation is if a teacher feels ill in a school building due to poor indoor air quality, then a ventilation system, or at least a properly sized portable air cleaner with filtration of particles and gases, if warranted, should be installed in the room to allow for more airflow. Finally, one training recommendation is to promote healthy apps and websites to teachers and to encourage app use not only for the CSDPs bought for school but also the products teachers purchase for use at home.

## Figures and Tables

**Table 1 ijerph-21-00211-t001:** Demographics of study participants.

Survey Questions	Total ^a^ (*n* = 205)	Total%	%Answered
School County region			
North	88	42.9%	51.8%
Central and Statewide	41	20.0%	24.1%
South	41	20.0%	24.1%
Missing	35	17.1%	
Race and Ethnicity ^b^:			
Middle Eastern/North African	2	1.0%	1.2%
**White ^d^**	**115**	**56.1%**	**67.3%**
Hispanic White	14	6.8%	8.2%
Non-Hispanic White	101	49.3%	59.1%
**Black ^d^**	**24**	**11.7%**	**14.0%**
Hispanic Black	4	2.0%	2.3%
Non-Hispanic Black	20	9.8%	11.7%
I prefer not to answer this question	23	11.2%	13.5%
Other	6	2.9%	3.5%
Missing	34	16.6%	
Gender Identity			
Male	67	32.7%	41.9%
Female	93	45.4%	58.1%
Missing/NA/I prefer not to answer	45	22.0%	
	Total ^c^ (*n* = 170)	Total%	%Answered
Number of Years Teaching in NJ (mean ± SD)	12.6 ± 7.4		
Missing	76		
Number of Years of Teaching Overall (mean ± SD)	13.2 ± 7.8		
Missing	76		
Birth Year (mean+-SD)	1976 ± 9.5		
Missing	78		
What is the highest education degree completed?			
Bachelor’s degree	25	14.7%	27.2%
Master’s degree	56	32.9%	60.9%
Doctoral degree	4	2.4%	4.3%
Other	7	4.1%	7.6%
IPNA/Missing	78	45.9%	
How many years of post-secondary education (after high school) have you completed? (mean ± SD)	6.2 ± 2.8		
Missing	77		

Note: ^a^ Includes questions/participants from all surveys ^b^ One participant identified as American Indian or Alaskan Native and one identified as Native Hawaiian or Other Asian–Pacific Islander. Race and ethnicity do not add to 100%, as some people may have selected more than one option. ^c^ Includes questions/participants from Survey 1 and Survey 2. ^d^ This is a summation of both those who identify as Hispanic and Non-Hispanic.

**Table 2 ijerph-21-00211-t002:** Reported locations where participants purchased cleaning, sanitizing, and disinfecting consumer products.

Survey Questions	Total ^f^ (*n* = 126)	Total%	%Answered
Where do you buy most of your cleaning products? (Pick all that apply)			
Grocery Stores ^a^	61	48.4%	58.7%
Big-Box Stores ^b^	30	23.8%	28.8%
By mail/Online ^c^	12	9.5%	11.5%
Hardware Stores ^d^	8	6.3%	7.7%
Pharmacies ^e^	8	6.3%	7.7%
I do not know; the school provides supplies	28	22.2%	26.9%
Missing	22	17.5%	
Where do you buy most of your sanitation products? (Pick all that apply)			
Grocery Stores	62	49.2%	62.0%
Big-Box Stores	28	22.2%	28.0%
By mail/Online	12	9.5%	12.0%
Hardware Stores	10	7.9%	10.0%
Pharmacies	9	7.1%	9.0%
I do not know; the school provides supplies	29	23.0%	29.0%
Missing	26	20.6%	
Where do you buy most of your disinfection products? (Pick all that apply)			
Grocery Stores	63	50.0%	60.6%
Big-Box Stores	30	23.8%	28.8%
By mail/Online	12	9.5%	11.5%
Hardware Stores	9	7.1%	8.7%
Pharmacies	7	5.6%	6.7%
I do not know; the school provides supplies	30	23.8%	28.8%
Missing	22	17.5%	

Note: ^a^ Included examples: Kings, ShopRite, Stop and Shop, Wegmans, etc. ^b^ Included examples: Costco, Sam’s Club, Target, Walmart, etc.^c^ Included example: Amazon. ^d^ Included examples: Ace Hardware, Home Depot, Lowes, etc. ^e^ Included examples: CVS, RiteAid, Walgreens, etc. ^f^ Includes questions/participants from Survey 1 and Follow-up Survey.

**Table 3 ijerph-21-00211-t003:** Reported locations where participants purchased cleaning, sanitizing, and disinfecting consumer products by age group.

Survey Questions	22–41 (*n* = 35) ^g^	%Answered	42+ (*n* = 57)	%Answered	Fisher’s Exact Test	Total (*n* = 92)	%Answered
Where do you buy most of your cleaning products? (Pick all that apply)							
Grocery Stores ^a^	21	60.0%	36	63.2%	0.83	57	62.0%
Big-Box Stores ^b^	15	42.9%	14	24.6%	0.10	29	31.5%
By mail/Online ^c^	9	25.7%	3	5.3%	0.01 *	12	13.0%
Hardware Stores ^d^	5	14.3%	2	3.5%	0.11	7	7.6%
Pharmacies ^e,f^	5	14.3%	2	3.5%	0.11	7	7.6%
I do not know; the school provides supplies	7	20.0%	16	28.1%	0.46	23	25.0%
Where do you buy most of your sanitation products? (Pick all that apply)							
Grocery Stores	20	57.1%	37	64.9%	0.51	57	62.0%
Big-Box Stores	14	40.0%	12	21.1%	0.06	26	28.3%
By mail/Online	9	25.7%	3	5.3%	0.01 *	12	13.0%
Hardware Stores	4	11.4%	4	7.0%	0.47	8	8.7%
I do not know; the school provides supplies	8	22.9%	16	28.1%	0.63	24	26.1%
Where do you buy most of your disinfection products? (Pick all that apply)							
Grocery Stores	21	60.0%	37	64.9%	0.66	58	63.0%
Big-Box Stores	15	42.9%	13	22.8%	0.06	28	30.4%
By mail/Online	9	25.7%	3	5.3%	0.01 *	12	13.0%
Hardware Stores	4	11.4%	4	7.0%	0.47	8	8.7%
I do not know; the school provides supplies	9	25.7%	16	28.1%	1.00	25	27.2%

Note: ^a^ Included examples: Kings, ShopRite, Stop and Shop, Wegmans, etc. ^b^ Included examples: Costco, Sam’s Club, Target, Walmart, etc. ^c^ Included example: Amazon. ^d^ Included examples: Ace Hardware, Home Depot, Lowes, etc. ^e^ Included examples: CVS, RiteAid, Walgreens, etc. ^f^ For sanitization and disinfection, six people under age 42 bought products in pharmacies and one person over 42 did. This was significant; however, due to the small number, the analysis is not valid. ^g^ Includes questions/participants from Survey 1 and Follow-up Survey. * *p*-value ≤ 0.01.

**Table 4 ijerph-21-00211-t004:** Participants’ self-reported practices of reading product ingredient lists.

Survey Questions	Total ^a^ (*n* = 126)	Total%	%Answered
When you buy _____ to use, how often do you read ingredients on the label?			
(a) cleaning products			
Very Often/Often	36	28.6%	36.7%
Sometimes	28	22.2%	28.6%
Not Often/Never	34	27.0%	34.7%
Missing/NA/I prefer not to answer	28	22.2%	
(b) sanitizing products			
Very Often/Often	36	28.6%	36.4%
Sometimes	31	24.6%	31.3%
Not Often/Never	32	25.4%	32.3%
Missing/NA/I prefer not to answer	27	21.4%	
(c) disinfection products			
Very Often/Often	37	29.4%	37.8%
Sometimes	28	22.2%	28.6%
Not Often/Never	33	26.2%	33.7%
Missing/NA/I prefer not to answer	28	22.2%	
When buying ____ to use, how often do you look for labels indicating the product is made with natural, non-toxic or eco-friendly ingredients?			
(a) cleaning products			
Very Often/Often	38	30.2%	38.8%
Sometimes	32	25.4%	32.7%
Not Often/Never	28	22.2%	28.6%
Missing/NA/I prefer not to answer	28	22.2%	
(b) sanitizing products			
Very Often/Often	38	30.2%	38.8%
Sometimes	30	23.8%	30.6%
Not Often/Never	30	23.8%	30.6%
Missing/NA/I prefer not to answer	28	22.2%	
(c) disinfection products			
Very Often/Often	38	30.2%	38.8%
Sometimes	30	23.8%	30.6%
Not Often/Never	30	23.8%	30.6%
Missing/NA/I prefer not to answer	28	22.2%	

Note: ^a^ Includes questions/participants from Survey 1 and Follow-up Survey.

**Table 5 ijerph-21-00211-t005:** Participants’ self-reported practices of reading product ingredient lists by race/ethnicity.

Survey Questions	Other (*n* = 34)	%Answered	Non-Hispanic White (*n* = 57)	%Answered	Fisher’s Exact Test	Total ^a^ (*n* = 91)	%Answered
When you buy _____ to use, how often do you read ingredients on the label?							
(a) cleaning products					<0.001 *		
Very Often/Often	13	39.4%	17	30.9%		30	34.1%
Sometimes	16	48.5%	10	18.2%		26	29.5%
Not Often/Never	4	12.1%	28	50.9%		32	36.4%
(b) sanitizing products					<0.001 *		
Very Often/Often	12	36.4%	18	32.1%		30	33.7%
Sometimes	17	51.5%	12	21.4%		29	32.6%
Not Often/Never	4	12.1%	26	46.4%		30	33.7%
(c) disinfection products					<0.001 *		
Very Often/Often	13	39.4%	18	32.7%		31	35.2%
Sometimes	16	48.5%	10	18.2%		26	29.5%
Not Often/Never	4	12.1%	27	49.1%		31	35.2%
When buying a ____ to use, how often do you look for labels indicating the product is made with natural, non-toxic or eco-friendly ingredients?							
(a) cleaning products					0.26		
Very Often/Often	12	36.4%	21	38.2%		33	37.5%
Sometimes	14	42.4%	15	27.3%		29	33.0%
Not Often/Never	7	21.2%	19	34.5%		26	29.5%
(b) sanitizing products					0.13		
Very Often/Often	12	36.4%	21	38.2%		33	37.5%
Sometimes	14	42.4%	13	23.6%		27	30.7%
Not Often/Never	7	21.2%	21	38.2%		28	31.8%
(c) disinfection products					0.13		
Very Often/Often	12	36.4%	21	38.2%		33	37.5%
Sometimes	14	42.4%	13	23.6%		27	30.7%
Not Often/Never	7	21.2%	21	38.2%		28	31.8%

Note: Please note that if a column does not add to the total *n*, this is because of missing responses, and these were not included in percentage calculation. ^a^ Includes questions/participants from Survey 1 and Follow-up Survey. * *p*-value ≤ 0.001.

**Table 6 ijerph-21-00211-t006:** Participants’ self-reported use of health-related apps or websites.

Survey Questions	Total ^a^ (*n* = 126)	Total%	%Answered
Before purchasing the _____, do you look it up on a healthy product app or website?			
(a) cleaning products			
Yes	16	12.7%	17.4%
No	76	60.3%	82.6%
Missing/NA/I prefer not to answer	34	27.0%	
(b) sanitizing products			
Yes	15	11.9%	16.5%
No	76	60.3%	83.5%
Missing/NA/I prefer not to answer	35	27.8%	
(c) disinfecting products			
Yes	17	13.5%	18.5%
No	75	59.5%	81.5%
Missing/NA/I prefer not to answer	34	27.0%	

Note: ^a^ Includes questions/participants from Survey 1 and Follow-up Survey.

**Table 7 ijerph-21-00211-t007:** Participants’ medical symptoms.

Survey Questions	Total ^a^ (*n* = 114)	Total%	%Answered
Have you ever had any of the following respiratory health problems after working in this school: runny nose, itchy/watery eyes, trouble breathing or headache?			
Yes	33	28.9%	31.1%
No	52	45.6%	49.1%
I’m not sure	21	18.4%	19.8%
Missing/NA/I prefer not to answer	8	7.0%	
	Total ^b^ (*n* = 33)	Total%	%Answered
Did you need to seek medical attention in response to any symptoms?			
Yes	9	27.3%	27.3%
No	24	72.7%	72.7%
Missing/NA/I prefer not to answer	0	0.0%	
	Total ^a^ (*n* = 114)	Total%	%Answered
If this is your 1st year at this school, did you experience these symptoms at your prior school or job?			
Yes	8	7.0%	15.7%
No	35	30.7%	68.6%
I’m not sure	8	7.0%	15.7%
Missing/NA/I prefer not to answer	63	55.3%	
Do any of these symptoms disappear if you are away from work more than a day?			
Yes	22	19.3%	37.3%
No	26	22.8%	44.1%
I’m not sure	11	9.6%	18.6%
Missing/NA/I prefer not to answer	55	48.2%	

Note: ^a^ Includes questions/participants from Survey 2 and Follow-up Survey. ^b^ Question only asked to those who said “yes” to “Have you ever had any of the following respiratory health problems after working in this school: runny nose, itchy/watery eyes, trouble breathing or headache?”.

## Data Availability

Data were obtained via three surveys named “Rutgers NJ Safe Schools Program New Work-Based Learning Supervising Teachers Cohort - 1st Survey (Parts I and II, and Demographics)”, “Rutgers NJ Safe Schools Program New Work-Based Learning Supervising Teachers Cohort - 2nd Survey [Follow-up on Safety, Health and Wellness (physical and mental)]”, and “Late 2022/2022-2023 School Year Follow-up (v.1 Aug2021) Rutgers NJ Safe Schools Program New Work-Based Learning Supervising Teachers Cohort – Survey”, which were delivered by New NJ teachers in the 2021-2022 and 2022-2023 school year.

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
