# Peer review of "Perspectives and Attitudes of Newer New Jersey High School Teachers towards Cleaning, Sanitizing, and Disinfecting Consumer Products Used in School Classrooms"

_ijerph, 2024, doi:10.3390/ijerph21020211_

Round 1
Reviewer 1 Report
Comments and Suggestions for Authors
1. In general, the article is well written, however, when citing literature, it is better to check again, especially citing articles that have not yet been published.
2. Please write the novelty in the article clearly, and mention the differences with previous research
3. Please state the importance of this study to answer the research problem
4. The titles in tables 2 and 3 are not written well, a table title is not a question sentence 5. Why are the numbers of participants in Tables 1, 2, 3 and 4 different? This makes it difficult for readers to understand the contents of the table 6. Why is there so much missing data in Tables 1 and 2? The author should have checked the questionnaire first before processing the data 7. Why is there still a citation in the conclusion (line 341)? Conclusions must be able to answer the main question of this study 8. In line 148 it says "(IRB protocol code: 2021001559". Does this require adding ")" at the end of that sentence? 9. In the manuscript it is not clearly stated whether prior informed consent was obtained from the participants before this study was carried out 10. In Table 5, why do you do the Fisher's Exact Test on each question? It is best to carry out statistical tests in accordance with the main research questions Comments on the Quality of English LanguageIt has been well written in English
Author Response
Reviewer 1
- In general, the article is well written, however, when citing literature, it is better to check again, especially citing articles that have not yet been published.
Thank you, we have checked all citations. Among those citations, the one which has not been published yet was only our group’s paper and it has been accepted as of 12/11/2023 (Reference 44), i.e., in press. We can provide MDPI updates as we receive them.
- Please write the novelty in the article clearly, and mention the differences with previous research.
Thank you for your idea. This text has been added to the introduction: “To our knowledge there have been few studies to date regarding schoolteachers and CSDPs. This is one of the first studies to specifically survey teachers regarding where they get their CSDPs, and their use of PPE while using these items.” (Lines 88-90).
- Please state the importance of this study to answer the research problem.
Thank you for the comment. This text has been added to the introduction: “The purpose of this research is to help ensure teacher and student S&H, given everyone needs to be aware of the products used in their learning/study, living and workspaces along with possible side effects of exposure to emissions of the products.” (Lines 91-93).
- The titles in tables 2 and 3 are not written well, a table title is not a question sentence.
We agreed to changes in the title for table 2 to “Reported Locations where Participants Purchased Cleaning, Sanitizing and Disinfecting Consumer Products” and for table 3 to “Reported Locations where Participants Purchased Cleaning, Sanitizing and Disinfecting Consumer Products, by Age Group” to comply with reviewer request.
- Why are the numbers of participants in Tables 1, 2, 3 and 4 different? This makes it difficult for readers to understand the contents of the table.
This paper had separate surveys with different questions. The numbers are different because some questions were only in certain surveys so the total number of participants who responded to each question is slightly different. We have ensured to specify which survey the questions are from appear in the footnotes of the tables.
- Why is there so much missing data in Tables 1 and 2? The author should have checked the questionnaire first before processing the data.
There were several participants who completed most of the surveys but did not finish it completely, including demographic questions. We chose to include them in the survey, but they are counted as missing.
- Why is there still a citation in the conclusion (line 341)? Conclusions must be able to answer the main question of this study.
We appreciate for this comment. The citation is relevant for this paper. Thus, this information has now been moved to the introduction to avoid citing it in the conclusion.
- In line 148 it says "(IRB protocol code: 2021001559". Does this require adding ")" at the end of that sentence?
Thank you for pointing this out. You are correct. This copy edit has been added.
- In the manuscript it is not clearly stated whether prior informed consent was obtained from the participants before this study was carried out.
The specifics of the online consent form were added to the paper in line 129. We also have sent the IRB approved form for consent without written documentation to MDPI; the IRB approved surveys had already been provided as a Supplemental file.
- In Table 5, why do you do the Fisher's Exact Test on each question? It is best to carry out statistical tests in accordance with the main research questions.
These questions asked are related to our main question, therefore the statistical test was conducted. We wanted to see how different demographics of teachers interacted with different cleaning, sanitizing, and disinfecting products. These survey questions asked how teachers self-reported to interact with the labels (looking for harmful ingredients).

Reviewer 2 Report
Comments and Suggestions for Authors
Thanks for inviting me to review this manuscript. Some major issues:
1. The study’s focus on New Jersey secondary school teachers might limit its generalisability to other regions or educational levels. While the paper mentions this limitation, it could benefit from a more in-depth discussion of how these findings could differ in other contexts or the potential for broader applicability.
2. The sample size of 205 participants, while reasonable, raises questions about its representativeness of the larger population of teachers. The paper could be strengthened by addressing how this sample size was determined and if it’s sufficient to draw robust conclusions.
3. More detailed methodological explanations would enhance the paper’s rigor. For instance, clarifying how the survey questions were developed and if they were validated would add to the credibility of the findings.
4. While the study presents a range of findings, there is room for deeper analysis and interpretation. For instance, exploring the reasons behind the reported disparities in PPE use among different demographic groups could provide more nuanced insights.
5. If a comparison could be made between the use of PPE by secondary school teachers and other groups, the conclusions of this paper might be more valuable. For example, there are some issues with the supply of PPE for Chinese migrant workers, but their employers try to mitigate the discomfort caused by the lack of PPE by offering higher wages (Liu et al., 2022). In this case, it seems that the use of PPE is primarily limited by supply, rather than the willingness or habit of using PPE.
6. The discussion section could be more expansive in linking the study’s findings to existing literature and theories. This would help situate the research within the broader field and highlight its unique contributions.
7. The paper could be more explicit in outlining practical implications and recommendations for school policy, teacher training, and health promotion. While there are some suggestions, a more detailed and actionable set of recommendations would be beneficial.
8. The paper briefly mentions the need for future research but could elaborate more on specific areas where further investigation is needed. For example, studying the long-term health impacts of cleaning product use among teachers or exploring similar practices in different educational settings could be interesting future directions.
9. The paper could explore the concept of perceived fairness in the context of workplace health and safety practices. This would align with broader themes of equity and workplace rights, especially in the context of pandemic responses.
10. Some relevant works: Goldman, 2020; Lan et al., 2021.
Overall, the paper provides a significant contribution to the understanding of paediatric healthcare accessibility in Texas, though there are areas where further depth and context could strengthen its impact.
Reference
Goldman, E. (2020). Exaggerated risk of transmission of COVID-19 by fomites. The Lancet Infectious Diseases, 20(8), 892-893.
Lan, F. Y., Suharlim, C., Kales, S. N., & Yang, J. (2021). Association between SARS-CoV-2 infection, exposure risk and mental health among a cohort of essential retail workers in the USA. Occupational and environmental medicine, 78(4), 237-243.
Liu, Q., Liu, Z., Kang, T., Zhu, L., & Zhao, P. (2022). Transport inequities through the lens of environmental racism: Rural-urban migrants under Covid-19. Transport policy, 122, 26-38.
Author Response
- The study’s focus on New Jersey secondary school teachers might limit its generalisability to other regions or educational levels. While the paper mentions this limitation, it could benefit from a more in-depth discussion of how these findings could differ in other contexts or the potential for broader applicability.
Thank you for this point. We expanded on our comment in the paper in the limitations: “In broader contexts, such as with a larger sample size research may reveal what is significant in this study may not continue to be and vice versa. We had a specific sample population so it is possible other populations such as teachers in general or even CTE teachers outside of NJ have different perceptions and attitudes regarding CSDPs. These data, however, can be used as comparison to other states or time frames regarding data for CTE teachers and their perceptions and attitudes regarding CSDPs.”(Lines 356-362).
- The sample size of 205 participants, while reasonable, raises questions about its representativeness of the larger population of teachers. The paper could be strengthened by addressing how this sample size was determined and if it’s sufficient to draw robust conclusions.
While 205 may seem a small sample size, in this study there were a total of 436 possible responses (163 for survey 1, 163 for survey 2, 110 for the follow up survey), so the question/answer-level response rate is about half or 47%. This is a representative sample of newer CTE teachers in NJ. We have added the following into our paper for clarity, to correspond to noted limitations in the discussion: “Our sample size estimate was 55 participants per survey, for a total sample size 165 among the three surveys. The estimated sample size for one survey was 55 with a 95% confidence level, a 5% margin of error, and a 4.5 % population proportion of CTE teachers in the general teacher population in the New York, NJ, and Pennsylvania metropolitan area in 2022.”(Lines 141-145).
We similarly responded to reviewer #3 on this topic as well.
- More detailed methodological explanations would enhance the paper’s rigor. For instance, clarifying how the survey questions were developed and if they were validated would add to the credibility of the findings.
Additional text was added to the methods section under survey details. Most of the questions in these surveys were adapted from previous New Jersey Safe School Program (NJSS) surveys or were crafted by the NJSS team (program director and staff) to provide questions deemed important to NJ teachers. We also cited common federal and state agency resources for questions, e.g., U.S. Census. The Rutgers IRB reviewed and approved the surveys used in this study for original and follow-up in 2021-2023.
- While the study presents a range of findings, there is room for deeper analysis and interpretation. For instance, exploring the reasons behind the reported disparities in PPE use among different demographic groups could provide more nuanced insights.
Due to the small sample size, we were not able to perform any further analysis regarding race and/or ethnicity. We added this as a recommendation for future studies.
- If a comparison could be made between the use of PPE by secondary school teachers and other groups, the conclusions of this paper might be more valuable. For example, there are some issues with the supply of PPE for Chinese migrant workers, but their employers try to mitigate the discomfort caused by the lack of PPE by offering higher wages (Liu et al., 2022). In this case, it seems that the use of PPE is primarily limited by supply, rather than the willingness or habit of using PPE.
Thank you for this resource. It has been included in the discussion. “A study in China reported there were issues providing PPE to migrant workers at the beginning of the pandemic. This study suggested potential ethnic dis-crimination present and the lack of PPE usage was not due to lack of want to use PPE but the lack of access to proper PPE . In this study, it is possible lack of access might be another reason NJ CTE teachers who do not identify as NHW do not use PPE as often as those who do.” (Lines 322-326).
- The discussion section could be more expansive in linking the study’s findings to existing literature and theories. This would help situate the research within the broader field and highlight its unique contributions.
We have expanded the discussion section to be more inclusive of existing literature. Please also see (with more references cited) response to comment 10.
- The paper could be more explicit in outlining practical implications and recommendations for school policy, teacher training, and health promotion. While there are some suggestions, a more detailed and actionable set of recommendations would be beneficial.
Thank you for this recommendation. We added four additional recommendations to the conclusion. The revised text now reads: “One policy we suggest is schools buy CSDPs for teachers and check these products are safe and have clean ingredients along with providing PPE for teachers to use as they clean their classrooms. Another related policy is to actually require teachers to use PPE while cleaning their classrooms including proper gloves, masks, and eyewear. A third policy recommendation is if a teacher feels ill in a school building due to poor indoor air quality, then a ventilation system or at least a properly sized portable air cleaner with filtration of particles and gases, if warranted, should be installed in the room to allow more airflow. Finally, one training recommendation is to promote the healthy apps and websites to teachers and encourage app use not only for CSDPs bought for the school but also the products teachers purchase for use at home.” (Lines 399-409).
- The paper briefly mentions the need for future research but could elaborate more on specific areas where further investigation is needed. For example, studying the long-term health impacts of cleaning product use among teachers or exploring similar practices in different educational settings could be interesting future directions.
Thank you for your suggestion. This idea along with others are now in the conclusion. The revised text now reads: “More studies need to be conducted in schools, which would allow researchers to look at the long-term impacts of using CSDPs in classrooms or if different classroom environments (typical classroom, labs, workshops, salons) experience different problems regarding CSDPs. This could also include an examination of if teacher behaviors change towards CSDP selection and use after learning about more about the dangerous ingredients (active and/or inactive chemicals) in CSDPs” (Lines 392-397).
- The paper could explore the concept of perceived fairness in the context of workplace health and safety practices. This would align with broader themes of equity and workplace rights, especially in the context of pandemic responses.
This paper does not focus on the perceived fairness in the context of workplace safety as we did not ask any questions about fairness in this survey.
- Some relevant works: Goldman, 2020; Lan et al., 2021.
Overall, the paper provides a significant contribution to the understanding of paediatric healthcare accessibility in Texas, though there are areas where further depth and context could strengthen its impact.
Reference
Goldman, E. (2020). Exaggerated risk of transmission of COVID-19 by fomites. The Lancet Infectious Diseases, 20(8), 892-893.
Lan, F. Y., Suharlim, C., Kales, S. N., & Yang, J. (2021). Association between SARS-CoV-2 infection, exposure risk and mental health among a cohort of essential retail workers in the USA. Occupational and environmental medicine, 78(4), 237-243.
Liu, Q., Liu, Z., Kang, T., Zhu, L., & Zhao, P. (2022). Transport inequities through the lens of environmental racism: Rural-urban migrants under Covid-19. Transport policy, 122, 26-38.
Thank you, these articles have been added to our paper.

Reviewer 3 Report
Comments and Suggestions for Authors
1. 'a total of 205 entries were received' Out of how many survey sent? Could there be bias if the response rate is low?
2. It's very misleading that the n of each survey are different. Readers might have an impression of the survey respondents' demographic characteristics after reading table 1, while table 2 results might be from those who did no even answer the 1st survey. It might be better to limit the study population to those who answered all the surveys even though the sample size will be reduced.
3. Some of the conclusions are not very reliable. For example, "The study also found about a third of participants experienced respiratory problems after working in their school, and about a third were relieved of symptoms, i.e., symptoms went away, after leaving the school premises. This could be consistent with the idea CSDPs are irritating to the respiratory tract, and after leaving the area in which these products were used, one can experience symptom relief". The authors did not compare these percentages to any comparator arms (e.g. school teacher not using CSDP or non teachers during the same time). It's really hard to reach this conclusion solely based on the survey results.
4. The authors concluded that"2022 and 2023 there was a decrease in PPE use in 278 schools.2022-2023 school year" However, in table 7, only 2020-2021 and 2021-2022 resutls were reported.
In summary, I suggest the authors limit study population to those who answered all the surveys, better organize the results with fewer tables, and be more conservative with conlcusions
Author Response
Reviewer 3:
- 'a total of 205 entries were received' Out of how many survey sent? Could there be bias if the response rate is low?
We similarly responded to reviewer #2 on this topic as well. There were 436 total opportunities for teachers to respond (163 for survey 1, 163 for survey 2, and 110 for the follow up survey). This is a 47% response rate. While this response rate is low and there is a possibility of undercoverage bias, we believe this study’s question-level response rate is sufficient. (Please note: The total opportunities for teachers to respond and sample size estimation is now in the text).
- It's very misleading that the n of each survey are different. Readers might have an impression of the survey respondents' demographic characteristics after reading table 1, while table 2 results might be from those who did not even answer the 1st survey. It might be better to limit the study population to those who answered all the surveys even though the sample size will be reduced.
In this study, we cannot determine who has taken all of the surveys as there is no identifiable information between the different surveys, therefore there is no way to determine which survey participants took all three surveys. Since there is no way to tell who consistently answered all of the survey questions, we chose to keep all of participant responses in the survey analysis. We had noted this within the discussion limitations.
- Some of the conclusions are not very reliable. For example, “The study also found about a third of participants experienced respiratory problems after working in their school, and about a third were relieved of symptoms, i.e., symptoms went away, after leaving the school premises. This could be consistent with the idea CSDPs are irritating to the respiratory tract, and after leaving the area in which these products were used, one can experience symptom relief”. The authors did not compare these percentages to any comparator arms (e.g. school teacher not using CSDP or non teachers during the same time). It’s really hard to reach this conclusion solely based on the survey results.
Thank you for your comment. All of our statements in the discussion are specific to this study, and merely suggestive not absolute. There are, to date, few data on teacher self-reported symptoms in the school building and if there is a difference in symptoms experienced after they leave the building. Please note we have added more information in the discussion to expand on and clarify this topic.
- The authors concluded that"2022 and 2023 there was a decrease in PPE use in 278 schools.2022-2023 school year" However, in table 7, only 2020-2021 and 2021-2022 results were reported.
Thank you for your comment. This statement means there was a difference in mask use between those who took the survey in the 2021-2022 school year and those who took the survey in 2022-2023 school year. We have amended the discussion to include the following “This study found that between those who took the survey in the 2021-2022 school year versus the 2022-2023 school year there was a decrease in mask use in schools.” (Lines 306-308). We also amended the paragraph in the results to provide more clarity.
In summary, I suggest the authors limit study population to those who answered all the surveys, better organize the results with fewer tables, and be more conservative with conclusions.
The original Table 7 was removed due to redundancy in the text and tables, in an effort to reduce number of tables; key points remain in text. However, we maintained other tables (Tables 1-6 and the new Table 7 [previously Table 8]) as they are important to provide clarity about the study to readers.

Round 2
Reviewer 2 Report
Comments and Suggestions for Authors
I'm satisfied with the revised manuscript. Thanks for sharing.